# Vision guides the twilight search for oviposition sites of the Asian tiger mosquito, *Aedes albopictus*

Si Yu Zhao[1☉], Hong Kai Liu[1☉], Zhen Sheng Xie[1☉], Yi Ming Wu[1], Pei Lin Wu[1], Tong Liu[1], Wen Qiang Yang[1], Jing Wu[1], JunYu Fu[1], Chun Mei Wang[1], Anthony A. James[2,3], Xiao-Guang Chen[1]*

1 Department of Pathogen Biology, Institute of Tropical Medicine, School of Public Health, Southern Medical University, Guangzhou, China, 2 Department of Microbiology & Molecular Genetics, University of California, Irvine, California, United States of America, 3 Department of Molecular Biology & Biochemistry, University of California, Irvine, California, United States of America

☉ These authors contributed equally to this work.
* xgchen@smu.edu.cn

**Data Availability Statement:** All data are available in the main text or the supplementary materials.

**Funding:** This work was supported by the Key-Area Research and Development Program of

## Abstract

### Background

Oviposition site selection is an important component of vector mosquito reproductive biology. The Asian Tiger mosquito, *Aedes albopictus*, is a major and important vector of arboviruses including Dengue. Previous studies documented the preference of gravid females for small, dark-colored water containers as oviposition sites, which they sought during the twilight period (dusk) of their locomotor activity. Vision plays an important role in this behavior, and factors such as the shape, size, and color of the container, light intensity, polarization, spectrum, and other visual cues guide the search for suitable oviposition sites, but the mechanistic factors driving this behavior are unclear.

### Methodology/Principal findings

We blindfolded adult female compound eyes and observed the effects of a lack of vision on the ability to discriminate and utilize preferred oviposition sites. Furthermore, the transcriptomes of blindfolded mosquitoes were screened to identify genes with vision-sensitive expression profiles and gene-editing was used to create non-functional mutations in two of them, *rhodopsin-like* (mutation designated 'rho-l$^{\triangle 807}$') and *kynurenine hydroxylase* (mutation designated 'kh$^w$'). Behavioral tests of both mutant and control strains revealed that the rho-l$^{\triangle 807}$ mutant mosquitoes had a significant decrease in their ability to search for preferred oviposition sites that correlated with a reduced ability to recognize long-wavelength red light. The kh$^w$ mutant mosquitoes also had a reduced ability to identify preferred oviposition sites that correlated with reductions in their ability to respond to variations in daily brightness and their ability to discriminate among different color options of the containers and background monochromatic light.

Guangdong Province (2022B1111030002) and National Natural Science Foundation of China (31830087, 82261128003/2022YFML1001), and the National Institutes of Health, USA (AI136850) to X-G.C. AAJ is a Donald Bren Professor at UCI. The funders had no role in study design, data collection and analysis, decision to publish, or preparation of the manuscript.

**Competing interests:** The authors have declared that no competing interests exist.

## Conclusions/Significance

This study underscores the importance of visual cues in the oviposition site selection behavior of adult female *Ae. albopictus*. We demonstrate that wild-type *rho-l* and *kh* gene products play a crucial role in this behavior, as mutants exhibit altered sensitivity or recognition of light intensity and substrate colors.

### Author summary

*Aedes albopictus* (Asian Tiger mosquito) relies on vision to select oviposition sites, favoring small, dark containers during twilight hours. To understand the underlying mechanisms, we blindfolded female mosquitoes and screened their transcriptomes, identifying *rho-l* and *kh* genes as potential vision-related factors. Gene editing experiments provided crucial insights into the roles of *rho-l* and *kh* in *Ae. albopictus* oviposition behavior. Mutants lacking functional *rho-l* (*rho-l$^{\triangle 807}$*) exhibited a significant decrease in their sensitivity to long-wavelength red light, leading to a reduced ability to locate preferred oviposition sites. Similarly, *kh$^w$* mutants displayed a decreased ability to respond to variations in daily brightness, as well as a reduced capacity to discriminate between different colors. This impaired color discrimination and sensitivity to brightness changes directly affected their ability to identify preferred oviposition sites. These findings highlight the crucial role of *rho-l* and *kh* in mosquito visual-mediated oviposition site selection, providing valuable insights into mosquito behavior and potential targets for vector control.

## Introduction

*Aedes albopictus* is an important vector of arboviruses, transmitting diseases such as dengue fever, Zika virus disease, and chikungunya fever, highlighting the urgent need for effective vector control strategies to mitigate these significant public health threats [1–3]. This mosquito also is one of the most invasive species in the world, having spread in less than 50 years from its original habitat in Asia to all parts of the globe except Antarctica. Different mosquitoes prefer different oviposition sites and *Aedes* sp. mosquitoes prefer to lay eggs in small containers of standing water (e.g. pots, cans, tires, [4–7]). Three-quarters of the growth and development of mosquitoes, including eggs, larvae, and pupae, are completed in the water in the oviposition site [8], and the adult insects disperse following emergence. Therefore, the oviposition sites are a crucial location for mosquito control. Various mosquito sensory organs play an important role in the search for oviposition sites [9,10]. Previous studies showed that mosquitoes use their olfactory capabilities to detect the presence of suitable oviposition sites that are characterized by the moisture level and specific odors emitted by microorganisms in the water [8,11–13].Additionally, visual cues, such as the shape, size, and color of the container, as well as light intensity, polarization, and spectrum, also play a crucial role in guiding mosquitoes to appropriate oviposition sites [5,14–21].The main visual organ of an adult mosquito is the compound eye, which is made up of many individual ommatidia. Each of these ommatidia can see a small part of the field of view individually, and the field of view of all the ommatidia is combined to form a complete visual image of the mosquito [22,23]. Ommatidia exhibit uniform morphology, each comprising a cornea, crystalline cone, retinal cells, pigment cells, and a basement membrane. Each ommatidium houses eight photoreceptor cells, designated R1-R8, with the

majority of opsins localized to these retinal cells [24]. Several opsins have been identified in mosquitoes [21,25,26], of which 10 have been identified in *Ae. albopictus* [27]. Different opsins can sense different wavelengths of light [28]. The photoreceptor pathway converts the light signal into electrical and chemical signals that are transmitted to the nerve cells in the brain [29,30].

Previous studies showed that *Aedes* sp. mosquitoes prefer to find oviposition sites at dusk during the twilight period of their locomotor activity, and prefer dark containers with small amounts of water [17,31,32].The specific visual cues and molecular mechanisms for this significant reproductive behavior are unclear, and in this study, we used black acrylic pigment applied to the compound eyes of *Ae. albopictus* (blindfolding) to block the visual pathway [27,33] to assess whether the oviposition site searching behavior was affected by vision. The transcriptome sequencing of samples from both the blindfolded compound eye treatment and the control groups identified differentially-expressed vision-related genes. This approach allows for further exploration of the role and mechanisms associated with the search for oviposition sites.

## Materials and methods

### Mosquito strains and breeding

The *Ae. albopictus* Foshan strain was donated by the Guangdong Provincial Center for Disease Control and Prevention and was raised in the laboratory under the following conditions: temperature 27±1°C, relative humidity (RH) 65±10%, light for 14 hours, darkness for 10 hours. The mosquitoes used in the experiment all emerged within 1 week. The blindfolded test used mosquitoes 2 days after the blood-feeding, while the search test used mosquitoes 3 days after the blood-feeding.

### Compound eye blindfolded test

Mosquito compound eyes are the principle organs that receive light, and as such, have an important role in both circadian and responsive activities [34,35]. Female mosquitoes were selected two days after a full bloodmeal, and the compound eyes were blindfolded with opaque, black water-soluble acrylic pigment (793, Marie's) with a brush (0000# drawing pen, Xie DeTang) [27,33]. The pigment has no pungent odor and has been used previously in *Ae. albopictus* [27]. The pigment was blended on a slide with deionized water and tested with a photometer (model OHSP-350C, Hangzhou Rainbow Spectrum) to ensure light transmittance of less than 0.1%. The mosquitoes were allowed to recover for 24 hours under blind conditions before proceeding with the experiments. Each mosquito was treated with 100 μL of soluble acrylic pigment.

### Compound eye blindfolded effect detection

In order to test the shading effect of blindfolding, shadow selection experiments were carried out between blindfolded female mosquitoes and control female mosquitoes three days after blood-feeding and mating. Female mosquitoes were anesthetized with carbon dioxide and 20 of them per trial are placed into a plexiglass cylindrical container measuring 600mm in length, with an outer diameter of 80mm, an inner diameter of 74mm, and a wall thickness of 3mm. The two ends of the container are fixed and closed with gauze, and the two sides are covered with black cloth. Organic containers are placed horizontally. After 1 hour, the number of female mosquitoes distributed on both sides are counted and the shadow index is calculated as the number of mosquitoes in black environment—number of mosquitoes in light

environment)/total number. The shadow index was used as a measurement with the higher the index, the stronger the orientation of the mosquitoes towards darkness. This experiment was conducted on both control and blindfolded groups with nine replicates each for statistical robustness.

## Mosquito oviposition site search experiments

Three-hole Mosq-ovitrap is a common container for monitoring mosquito density in breeding sites, which consists of a transparent cylindrical plastic jar with a concave bottom and a black top with three tapered holes [36]. Two three-hole Mosq-ovitrap were placed diagonally in a 1 m$^3$ mosquito cage with temperature maintained at 28~30˚ C and humidity at 40~50%. A total of 100ml of dechlorinated water was added to one and the other was left empty.

Twenty gravid mosquitoes were put into mosquito cages after 3 days after a bloodmeal, and the number of gravid mosquitoes in the three-hole Mosq-ovitrap was recorded six hours later (12:00pm-18:00pm). The search index is the number of mosquitoes in the water-filled three-hole Mosq-ovitrap /total mosquito numbers. The search indices of compound eye blindfolded and control mosquitoes were measured and the experiments repeated six times for each group.

## Transcriptome sequencing

We compared the expression of visual genes associated with the phototransduction pathway during oviposition site selection in mosquitoes with and without blindfolded compound eyes. Test group: One hundred of female mosquitoes were blindfolded on their compound eyes 48 hours after blood feeding, then released into a 1m$^3$ mosquito net. Two three-hole oviposition traps were placed diagonally in the net, one containing 100 ml of deionized water and the other empty. Heads of female mosquitoes that flew into the three-hole Mosq-ovitrap with water were collected and stored in TRIZOL, then 30 individuals were pooled for sequencing. Control group: One hundred of female mosquitoes without compound eye blindfolded treatment were kept in the same rearing environment and batch as the test group. They were released into a 1m$^3$ mosquito net 72 hours after blood feeding. Two three-hole oviposition traps were placed diagonally in the net, one containing 100 ml of deionized water and the other empty. Heads of female mosquitoes that flew into the three-hole Mosq-ovitrap with water were collected and stored in TRIZOL, then 30 individuals were pooled for sequencing. A total of three biological replicates. The samples were sent to Beijing Novogene Sequencing Company for transcriptome sequencing. Transcriptome sequencing was repeated three times in both experimental and control groups. RNA integrity and total volume was measured accurately using the Agilent 2100 bioanalyzer. Qualified libraries were pooled and Illumina sequencing carried out to generate 150 base-paired (bp) end read. Raw data was filtered to remove reads with adapters, those containing N (indicating that base information cannot be determined) and low-quality reads (Qphred<-20 base number accounts for more than 50% of the entire read length), and at the same time, the Q20, Q30 and GC content of the clean data were calculated. All subsequent analyses are conducted with the high-quality clean data. This filtration procedure resulted in clean reads suitable for subsequent bioinformatics analysis (see: https://ftp.ncbi.nlm.nih.gov/genomes/all/GCF/006/496/715/GCF_006496715.2_Aalbo_primary.1/).

Differential expression analysis between the two comparison groups was performed using the DESeq2 software (version 1.20.0). DESeq2 provides statistical procedures to identify differentially expressed genes in digital gene expression data using a model based on the negative binomial distribution. The resulting p-values were adjusted using the Benjamini-Hochberg method to control the false discovery rate. Genes with an adjusted $P<$ 0.05 were considered

differentially expressed as identified by DESeq2. Visually relevant genes were identified among the differentially accumulated transcripts based on their functional annotation and gene description. The NCBI Blastn tool was used to validate the accuracy of gene sequencing and naming. Here we focused on the down regulated genes and identified 6 genes related to opsin synthesis in the head: *opsin-1* (LOC10942179), *opsin-2* (LOC109399710), *opsin-3-like* (LOC115260659), *opsin-1-like* (LOC109422560), *rhodopsin-like* (LOC115254593), *kynurenine 3-monooxygenase* (LOC109403230). We defined genes with a fold change (FC) of more than 1-fold (log2FC$\geq$0) and a *P* < 0.05 as significantly differentially expressed genes, and the remaining list of differentially-expressed genes was included in the S4 Table.

## Reverse Transcription-Quantitative Real-time PCR detection of *rho-l* and *kh* gene transcripts

Two-day post-blood-fed mosquitoes were treated with the acrylic pigment and allowed to recover for 24 hours. The head was removed and pools of five represented one biological replicate. A total of three biological replicates were performed. Control mosquitoes were treated similarly except for the application of the acrylic paint. The same sample sizes and replicates were collected and analyzed. Although there are five genes associated with opsin in the head transcriptome, we selected the *rhodopsin-like* gene with the differential fold, which is not a gene in opsin 1–10 identified in *Ae. albopictus* [27]. Under normal circumstances, opsins in compound eyes can bind with G proteins and initiate downstream phototransduction pathways [37,38].

We have discovered that the rhodopsin-like gene has an amino acid structure similar to that of opsins, leading us to hypothesize that this gene also has a photosensitive function, thereby influencing mosquitoes' behavior in seeking breeding sites through visual pathways. Meanwhile, Kynurenine 3-monooxygenase is an expressed product of KMO (LOC109403230), a member of the kynurenine metabolic pathway and has been shown to be an important monooxygenase in the tryptophan pathway [39]. Deletion mutations in the KMO gene and the resulting phenotypic changes were first identified in *Aedes aegypti*. The gene encoding this enzyme is called kynurenine hydroxylase (*kh*) and the mutant allele producing white eyes is called *kh^w* [40]. kynurenine-3-monooxygenase is associated with tryptophan metabolism and synthesis in pigment cells, which regulate the amount of light entering and shield stray light [41,42], and are also analyzed by RT-qPCR. We anticipate that the decreased level of light entering the compound eye will result in a decrease in the accumulation of transcripts from light-sensitive genes [26].

The dissected tissues were transferred into 1.5 ml RNase-free EP tubes containing 50 μL of Trizol (Ambion, Life Technologies, Carlsbad, CA, United States). Total RNA was extracted according to the manufacturer's protocol with Trizol reagent and dissolved in 20 μL of RNase-free water or DEPC-treated water. RNA was used for quantitative analysis using a Nanodrop 2000 (Thermo Technologies, the USA), and the concentration and 260/280 ratio were recorded. A 4ul volume of RNA was retained, and the rest was stored at -80˚C. First-strand cDNA was synthesized from DNaseI-treated total RNA using an oligo-(dT) primer and GoScript Reverse Transcription System kit (Promega Corporation, Madison WI, USA). RT-qPCR was performed at 95˚C for 10s and 60˚C for 15s, 72˚C, for 20s with a total of 40 cycles. Relative expression levels of *kh* (LOC109403230) and *rho-l* (LOC115254593) transcripts were calculated as $2^{-\Delta\Delta Ct}$ and normalized to the *actin-5C* reference gene (LOC109405344) transcript levels in the same cDNA samples. The primers used for gene expression analysis are shown in S2 Table.

### CRISPR/Cas9 editing experiments of *rho-l* and *kh* genes

Gene editing is a powerful tool for functional analysis. Following identification of target genes through transcriptome data screening and validation with RT-qPCR, we performed knockout experiments for the *rho-l* and *kh* genes. A dual sgRNA knockout strategy was used to produce a large deletion in the genome using the online tool, CRISPOR (http://crispor.tefor.net) to identify appropriate guide RNAs (sgRNAs) specifically targeting 20 bp of the site.

The injection mixes contained 300 ng/ul Cas9 Protein (Thermos Fisher Scientific), 100 ng/ul purified sgRNA1 and 100 ng/ul purified sgRNA2 added to RNase-free water. Two sgRNAs targeting the first and second exons were included with the Cas9 endonuclease in an injection mixture and this was microinjected into embryos at 1h after egg deposition. The injection mixtures were incubated for 15 min at 37°C to reconstitute active ribonucleoproteins (RNPs) and then injected into phenotypically wild-type Foshan strain embryos using described procedures [43–45]. Interbreeding allowed the recovery of a homozygous line of a single mutant type by the $G_3$ generation. The nucleotide sequence of the deletion mutation was obtained from a commercial sequencing company. To initiate the process, retrieve the protein sequence of *rho-l*. Subsequently, access the Swiss-Model website(https://swissmodel.expasy.org/) and select the automated modeling option to automatically generate a protein structure model. The amino acid structure of *rho-l* was mapped based on the results of the protein model.

The *kh* gene CRISPR/Cas9 edited strain, $kh^w$ is derived from a previously-constructed gene knockout strain [44].

### Color bidirectional selection experiments

We were intrigued by the observation that *Ae. albopictus* mosquitoes preferentially seek out black oviposition containers during the twilight hours. Based on the characteristics of twilight and the mosquitoes' preference for black oviposition containers, we conducted an oviposition preference experiment comparing black containers with those of other colors. The two egg-laying cups (black or white, black or blue, black or green, black or yellow, black or red) were placed in a 1m³ mosquito cage and 100ml of dechlorinated water was added to each. Ten gravid females were released into each cage, and the number of eggs deposited into each oviposition cup was recorded 24 hours later. Oviposition activity index (OVI) = (Number of eggs in non-black control cups–Number of eggs in black cups) / Total number of eggs laid in the experiment. The calculation of each data is based on the comparison of the number of eggs in black or other-colored cups in one experiment. The more the oviposition activity index tends to -1, the more the gravid mosquito prefer the black oviposition cup. The oviposition activity index was calculated for a total of 6–8 repeats. For a clear visual representation of the experimental process, a flowchart is presented in S3 Fig.

### Mosquito attraction to different brightness levels of light

Brightness is not constant throughout the day and the light intensities at noon, dusk and night can vary greatly [46]. Twilight is characterized by varying light intensities. To investigate this, we conducted attraction experiments under three different light conditions. A three-hole Mosq-ovitrap with 100ml dechlorinated water was placed in a 23cm×20cm×20cm mosquito cage, and the brightness of an LED lamp above the mosquito cage was adjusted to 0 lux (dark), 130 lux (twilight) or 1600 lux (daylight) [47]. Brightness is measured by an illuminance measuring instrument (Model OHSP-350C, Hangzhou Rainbow Spectrum). Thirty 3-day post-blooded females were placed into the cage, and the numbers of mosquitoes flying into the three-hole oviposition trap were recorded one hour later. A total of nine repetitions were performed.

## Mosquito attraction to different monochromatic light

Behavioral experiments with different light intensities (illuminance), colors of oviposition containers, and monochromatic light exposure were carried out to determine the role of the wild-type *rho-l* gene product in oviposition site choice. A three-hole oviposition trap containing 100ml of deionized water was placed in a 23cm×20cm×20cm mosquito cage under blue (460nm), green (520nm) or red (630nm) light wavelengths. Thirty 3-day post-blooded females were put into the mosquito cage, and the numbers of mosquitoes flying into the three-hole oviposition trap were recorded one hour later. A total of six replicates were performed for each group.

## Locomotor Activity analysis of the *rho-l* mutant strain of *Aedes albopictus*

*Ae. albopictus* wild-type strain and *rho-l*$^{\Delta 807}$ strain were reared in a standardized mosquito rearing room until adult mosquitoes, anaesthetized with $CO_2$, and then fully mated females of 3–5 days old were selected and placed on a Locomotor Activity Monitoring System (TriKinetics, LAM25) for activity monitoring. Activity monitoring is a routine method of assessing the capacity of insects, and *Drosophila* '(Diptera)' is more commonly used, measured by the number of flights through the infrared detection window. At present, *Drosophila* activity monitor has been widely used in mosquito activity monitoring [48–51].

The activity monitor was placed in a climatic chamber with the following conditions: temperature 27±1˚C, humidity 75%, light cycle LD = 12h:12h, with the light on time ZT0 (9:00am) and off time ZT12 (21:00pm), and monitored continuously for 7 days under the LD light conditions. The first three days of the LD phase are an adaptation period, while the subsequent four days are the formal monitoring period.

## Statistical analysis

All statistical analyses were performed using SPSS version 21.0 (IBM SPSS Statistics). All data were tested first to determine if they followed a normal distribution using a Shapiro-Wilk normality test ($\alpha = 0.05$). The independent sample t-test was used for comparison of the search index, mRNA relative expression levels of *rho-l* and *kh* genes, oviposition activity indices, mosquito numbers trapped by different brightness, mosquito numbers trapped by different monochromic light regimens, activity monitoring. One-way ANOVA was used to compare oviposition indices across multiple groups. A value of $P<0.05$ was considered to be statistically significant. The relevant statistics are already available in S5 Table.

# Results

## The adult compound eye of female *Aedes albopictus* plays an important role during the search for small water containers as oviposition sites

We blindfolded adult female mosquitoes with acrylic pigment to test if vision affects their ability to search oviposition sites (Fig 1A).

The shadow index of control mosquitoes was 0.31±0.04 while that of blindfolded insects, -0.21±0.09, was significantly lower (S1 Fig, t = 5.06, *df* = 16, $P<0.05$), indicating that the pigment application effectively blocked or lowered the level of incoming light and hindered light perception.

We built a model to record the search index (Fig 1B). The results showed that the search index was 0.51±0.04 for the compound eye blindfolded group and 0.70±0.03 for the mosquitoes in the normal compound eye group, and the difference between the two was statistically significant (Fig 1B, t = 3.215, *df* = 10, $P<0.01$), indicating that compound eye blindfolded

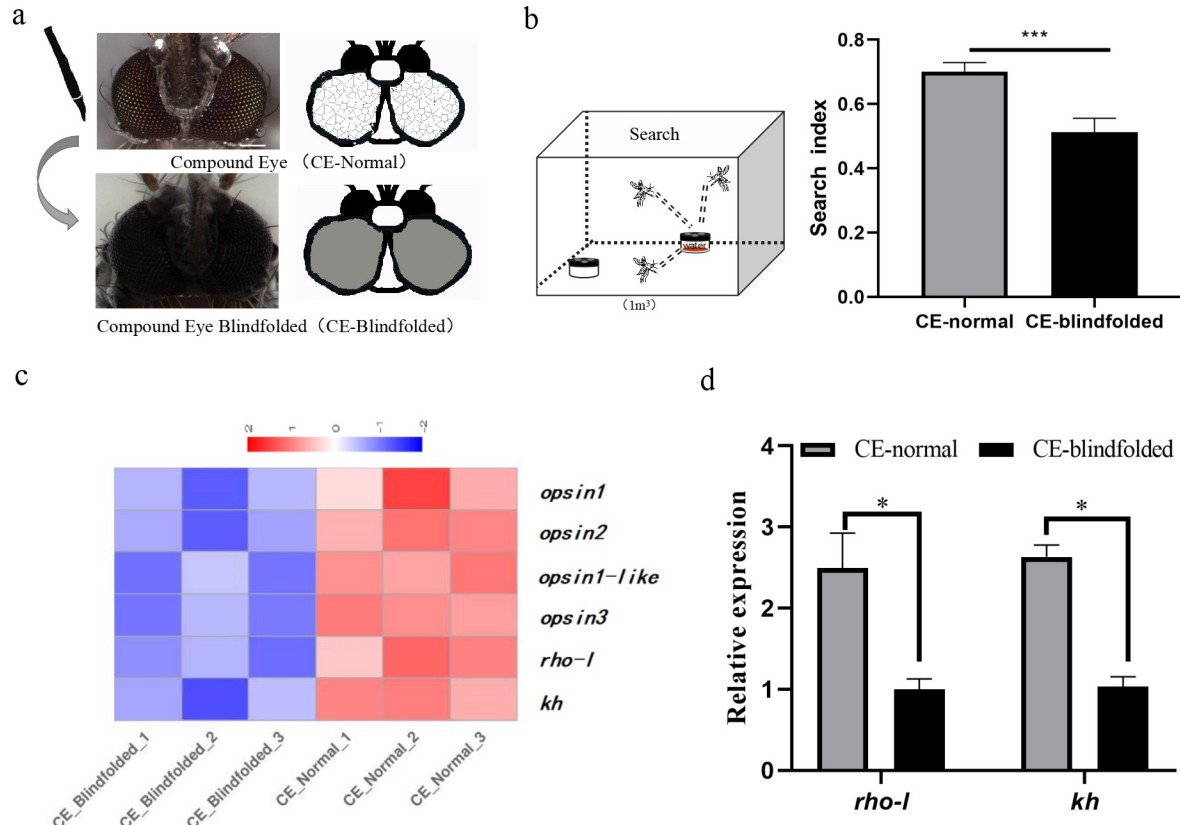

**Fig 1. Compound eye blindfolding and its effect on the search for oviposition sites. (a)** Compound eye blindfolding and pattern diagram. The top images are a photo (left) and schematic representation (right) of the control eyes and the bottom images show a blindfolded specimen. The scale size is 0.1mm. **(b)** Two three-hole egg traps are placed inside the 1m³ tent, one with water and one blank. Effect of compound eye blindfolding on oviposition sites of gravid mosquitoes (t-test, ***, *P*<0.001). **(c)** Compound eye blindfolding results in differential expression of vision-related genes. **(d)** RT-qPCR detection of *rho-l* and *kh* gene transcripts after blindfolding compound eyes (t-test, *, *P*<0.05). Data in (b) and (d) are represented as means ± SEM. CE- Normal means compound eye normal, and CE-blindfolded means compound eye blindfolded.

mosquitoes had a reduced ability to search for small pools of water in the container type compared to normal mosquitoes and suggesting that compound eye blindfolding affected the ability of *Ae. albopictus* to search for oviposition sites.

A comparative analysis of the transcriptomes of the blindfolded and control groups showed that six genes related to vision were regulated differentially with five showing reduced and one showing increased accumulation using |log2FoldChange|≥0 and *p* value <0.05 as the screening criteria (Fig 1C). The rhodopsin-encoding gene, *rho-l* (gene ID 115254593), showed significant reductions in transcript accumulation (S4 Table; log2FC = -1.81) compared with controls (Negative binomial distribution, *P* <0.001). Furthermore, RT-qPCR detection also showed that the relative expression of *rho-l* transcripts decreased significantly with blindfolding (Fig 1D, t = 3.345, *df* = 4, *P* <0.001), supporting the conclusion that the gene may be involved in compound eye-mediated light detection.

Transcriptomic data and RT-qPCR analyses show that the accumulation of *kh* transcripts decreased significantly in blindfolded mosquitoes compared to controls (S4 Table and Fig 1C and 1D). The combined results support the conclusion that the *rho-l* and *kh* genes are responsive to light levels and we posit that they may have a role in the compound eye in the search for oviposition sites.

## The *rho-l* gene product is essential for detecting long-wavelength red light during oviposition site searches

The *rho-l* gene transcript was reduced the most of all vision-related gene products in blind-folded mosquitoes compared to controls in the RNA-seq analyses (S4 Table). RT-qPCR results confirmed this reduction in accumulation and therefore we used CRISPR/Cas9 gene editing technology to generate mutant mosquitoes with an ablation of the *rho-l* gene and analyze its function. The *rho-l* has two exons and a single intron and produces a transcript with a total length of 1632 base-pairs (bp) (Fig 2A).

Based on the results of this analysis, we created a structural diagram of the amino acids in *rho-l* (Fig 2B). *Rho-l* mutations were found in the resulting $G_0$ adult mosquitoes, with four mutant adults identified from a total of 340 injected embryos (S3 Table). A homozygous mutant line with an 807 bp deletion was identified in the $G_3$ generation (Fig 2A and S1 Table). The *rho-l*$^{\triangle 807}$ product has only two of the seven transmembrane domains resulting in deletions of the conserved retinal binding and G-protein coupling sites (Fig 2B) [52]. Analysis of the expression profiles of mutant mosquito larvae, pupae, and 3-day post-eclosion adults showed that the *rho-l*$^{\triangle 807}$ mRNA was not expressed (Fig 2C). As an important opsin, the expression level of this gene in controls increased gradually from larval to pupal stage to adult stage, indicating that the main physiological function stage of this gene was required in the adult stage. Behavioral experiments showed that the search index of wild-type gravid

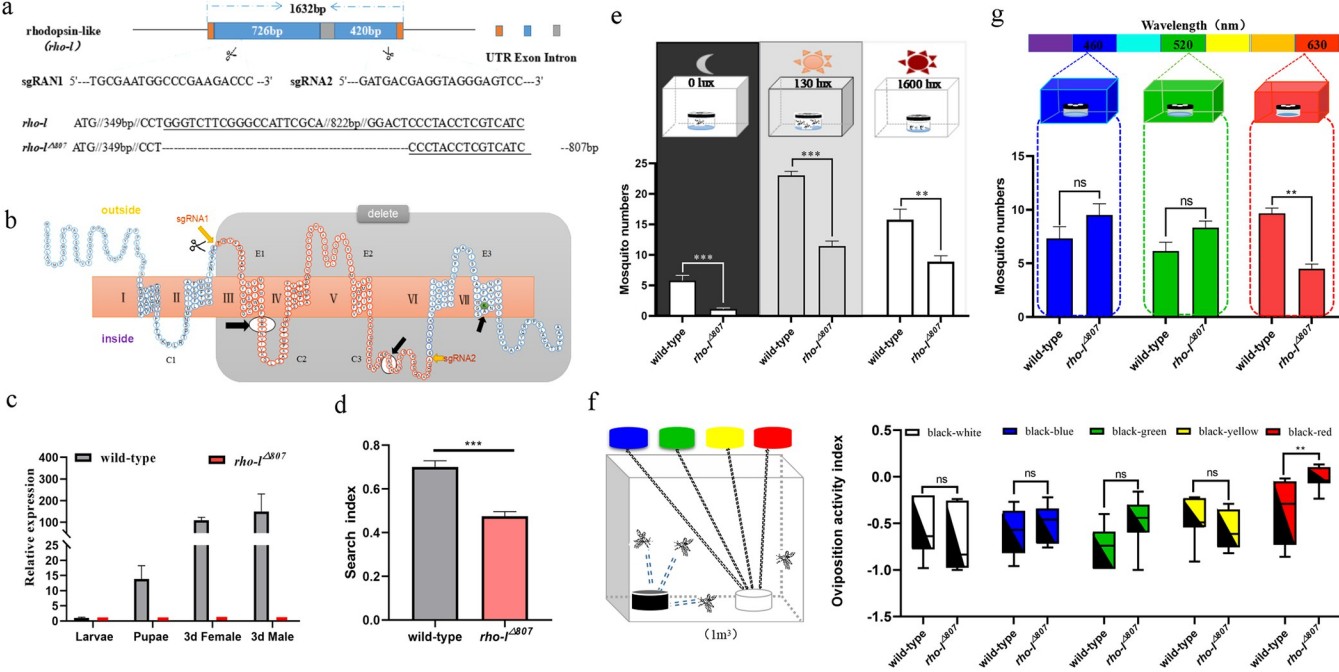

**Fig 2. Functional analysis of a *rho-l* ablation mutation.** (a) Schematic diagram of the *rho-l* gene mutation. (b) *rho-l* protein structure: "outside" means extracellular, "inside" means intracellular, "delete" means the transmembrane domain is deleted. (c) Comparison of mRNA accumulation levels between wild-type control and *rho-l*$^{\triangle 807}$ mutant lines during development. "3d Female" represents the female mosquito three days after emerging, "3d Male" represents male mosquitoes three days after emerging. (d) Comparison of search indices between wild-type control and *rho-l*$^{\triangle 807}$ (***$P < 0.001$). (e) Comparison between the numbers of wild type and *rho-l*$^{\triangle 807}$ mosquitoes in the traps under the brightness conditions of 0, 130 and 1600 lux (**$P < 0.01$ and ***$P < 0.001$, n = 9). (f) Bidirectional selection on two-color oviposition cups. Comparison of egg-inducing index between wild-type control and *rho-l*$^{\triangle 807}$ on black-white, black-green, black-yellow, black-blue, and black-red oviposition cups (**$P < 0.01$, NS. $P > 0.05$, n = 6). Oviposition activity index = (Number of eggs in non-black control cups–Number of eggs in black cups) / Total number of eggs laid in the experiment. (g) Comparison of the number of by wild type control and *rho-l*$^{\triangle 807}$ mosquitoes attracted under red (center wavelength 630nm), blue (center wavelength 460nm) and green (center wavelength 520nm) light (**$P < 0.01$ and NS. $P > 0.05$, n = 6). Data in (d), (e), (f) and (g) are represented as means ± SEM, and Student's t test was used (NS. $P > 0.05$, *, $P < 0.05$, **, $P < 0.01$, ***, $P < 0.001$).

mosquitoes flying into the three-hole Mosq-ovitrap was 0.70±0.02, while that of $rho\text{-}l^{\triangle 807}$ mutant strain, 0.47±0.02, was significantly lower (Fig 2D, t = 6.26, $df$ = 10, $P$ <0.001). The reduced ability of the $rho\text{-}l^{\triangle 807}$ mutant strain to find oviposition sites supports the conclusion that the wild-type product plays an important role in the visual process of oviposition site searching.

Behavioral experiments with different light intensities (illuminance), colors of oviposition containers, and monochromatic light exposure were carried out to determine the role of the wild-type $rho\text{-}l$ gene product in oviposition site. The trapping results of control wild-type gravid mosquitoes differed under the three conditions. The maximum numbers of mosquitoes caught at 0 (night), 130 (dusk), and 1600 lux (day) were 5.10±0.92, 23.00±0.71 and 16.73±1.62 respectively (Fig 2E). *Ae. albopictus* females require a certain illuminance to search for oviposition sites, and the 130 lux near the dusk stage is optimal. The trapping numbers, 1.11±0.20, 11.44±0.81, and 8.89±0.96, of the $rho\text{-}l^{\triangle 807}$ strain at 0, 130, and 1600 lux, respectively, were significantly lower than those of the wild type (Fig 2E, t = 4.22, $df$ = 9.8, $P$<0.001; t = 10.68, $df$ = 16, $P$<0.001; t = 4.16, $df$ = 16; $P$<0.01), indicating that the product of the $rho\text{-}l$ gene has a role in the perception of light intensity and deletion of the gene will affect recognition of light intensity and the ensuing search for oviposition sites. Regarding the decreased searching ability of the $rho\text{-}l$ strain in a dark environment, one reason is the impact on its activity. We monitored the activity of the $rho\text{-}l^{\triangle 807}$ strain under LD light conditions (S2 Fig). *Ae. albopictus* wild-type and $rho\text{-}l^{\triangle 807}$ strain both showed a clear bimodal activity pattern with a morning peak around lights on and an evening peak around lights off (S2A Fig). Compared with the wild-type, the activity of $rho\text{-}l^{\triangle 807}$ strain was significantly weakened during both daytime and nighttime (S2B Fig, t = 2.91, $df$ = 6, $P$<0.05; t = 5.3, $df$ = 6, $P$<0.01), and its flight activity was indeed affected. Another potential reason for the reduced searching ability could be that the $rho\text{-}l^{\triangle 807}$ strain relies on light recognition, and it may need a longer time to adapt to changes in the light-dark environment. Our experimental trapping period was only 1 hour, which might not have been sufficient for this adaptation. These data support the conclusion that $rho\text{-}l$ gene products are one of the visual determinants for *Ae. albopictus* oviposition site searching.

The oviposition activity index of wild-type controls was 0.53±0.11, -0.60±0.08, -0.74±0.08, -0.47±0.08 and -0.38±0.33 in black-white, black-blue, black-green, black-yellow and black-red choices, respectively (Fig 2F). While the results show that the wild-type mosquitoes all preferred black oviposition containers, the differences were not statistically significant (Fig 2F, F = 2.62, $P$ >0.05). The black and red results here do not show that the $rho\text{-}l$ strain of mosquito has an increased tendency towards red containers, but rather an inability to recognize red containers. While there were no statistically-significant differences in the $rho\text{-}l^{\triangle 807}$ oviposition numbers for the black-white, black-blue, black-green, and black-yellow choices compared with the controls (Fig 2F, t = 0.85, $df$ = 11; t = 0.38, $df$ = 13; t = -1.85, $df$ = 12; t = 0.86, $df$ = 11; $P$ >0.05), the oviposition index of the black-red choice was significantly-different from that of the wild type (Fig 2F, t = -2.61, $df$ = 12, $P$ <0.01). These data support the conclusion that wild-type $rho\text{-}l$ gene products have a significant role in the recognition of red containers, and deletion of the gene will significantly affect the oviposition site choice between red containers and the surrounding environment, thus affecting the identification and localization of oviposition sites.

We used red, green, and blue LEDs to study the role of light spectra in oviposition site choice and the phenotypic differences in the $rho\text{-}l^{\triangle 807}$ strain. The mean numbers of gravid mosquitoes trapped with blue light were 7.33±1.08 for the control wild-type and 9.5±1.06 for the mutant strains and these were not statistically significant (Fig 2G; t = -1.43, $df$ = 10, $P$ = 0.183). Similarly, no significant differences were found under green light conditions with the mean numbers of 6.14±0.82 and 8.33±0.61 for control and mutant mosquitoes, respectively

([Fig 2G](); t = -2.06, *df* = 10, *P* = 0.064). In contrast, the mean numbers of mosquitoes trapped were significantly different in the red-light environment with 9.67±0.49 and 4.50±0.43 for control and mutant mosquitoes, respectively ([Fig 2G](); t = 7.9, *df* = 10, *P* <0.001).

## The *kh* gene products have a critical role in oviposition site searching by regulating the quantity and quality of light

The pigment in the cells regulates the amount of light entering and forms different color patterns. Three primary colors usually make up a variety of patterns. The *kh* gene plays a crucial role in pigment synthesis by catalyzing a late stage in production of ommochromes [41]. The comparative analysis of the transcriptomes of the blindfolded and control groups revealed a lower level of accumulation of *kh* transcription products in the experimental group and this was confirmed further by RT-qPCR results ([Fig 1E]()). This supports the hypothesis that *kh* gene products might be involved in the light-pathway reaction mediated by the compound eye.

A *kh* gene knockout strain was obtained from previous work ([Fig 3A]()) [44], and like many in the broader family of mosquitoes, homozygous $kh^w$ have a white-eye phenotype ([Fig 3B]()).

Behavioral experiments showed that the search index of the $kh^w$ strain was 0.13±0.02, significantly lower than that of wild mosquitoes ([Fig 3C](); 0.70±0.02, t = 16.37, *df* = 10, *P* <0.001), indicating that the *kh* gene knockout affected oviposition site searching behavior. The mean trapping numbers of the $kh^w$ strain under 0, 130, and 1600 lux were 1±0.86, 1.22±0.97 and 0.88 ±0.78, respectively, with controls having the previously reported respective values of 5.10 ±0.92, 23.00±0.71 and 16.73±1.54 ([Fig 3D]()), with the latter two values being statistically significance ([Fig 3D](), t = 4.23, *df* = 10.73, *P*<0.001; t = 28.0, *df* = 16, *P*<0.001; t = 9.66, *df* = 10.52, *P*<0.001). Consequently, the results could be similar to those seen in the *rho-l* mutant strain.

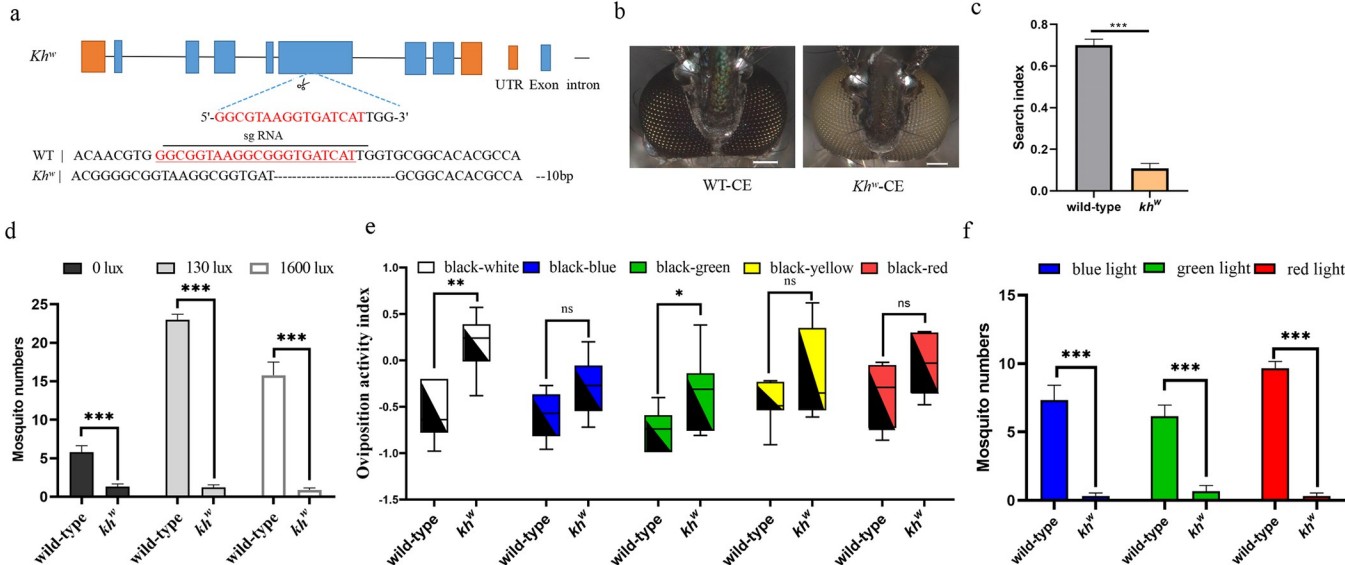

**Fig 3. Functional analysis of a $kh^w$ mutant mosquitoes. (a)** Schematic diagram of the mutant, $kh^w$, gene knockout structure [44]. **(b)** Phenotype comparison of $kh^w$ knockout lines and wild-type compound eyes. The scale size is 0.1mm. **(c)** Comparison of $kh^w$ strains and wild-type search index (***$P$ < 0.001, n = 6). **(d)** Comparisons of the numbers of $kh^w$ and wild-type controls recorded under the brightness conditions of 0 lux, 130 lux, 1600 lux (**$P$ < 0.01 and ***$P$ < 0.001, n = 9). **(e)** Comparisons of bidirectional egg-trapping indices of gravid wild-type and $kh^w$ mutants in black-white, black-green, black-yellow, black-blue, and black-red oviposition cups. Oviposition activity index = (Number of eggs in non-black control cups–Number of eggs in black cups) / Total number of eggs laid in the experiment. (*$P$ < 0.05, **$P$ < 0.01, and NS. $P$>0.05, n = 6–8). **(f)** Comparison of the numbers of wild-type control and $kh^w$ mosquitoes attracted by red (center wavelength 630nm), blue (center wavelength 460nm), and green (center wavelength 520nm) lights (***$P$ < 0. 001.n = 6–7). Data in (c), (d), (e) and (f) are represented as means ± SEM, and Student's t test was used (NS. $P$>0.05, *$P$ < 0.05, **$P$<0.01, and ***$P$<0.001).

These results support the interpretation that the deletion of the *kh* gene and loss of its products affected the search and location of oviposition sites. The results of testing the *kh$^w$* strain in the bidirectional selection of different color oviposition cups showed that the oviposition indices of the black-white and black-green groups were 0.18±0.13 and -0.35±0.16, respectively, and the difference was statistically significant compared with control wild-type mosquitoes (Fig 3E, t = -4.08, *df* = 11, *P*<0.01; t = -2.20, *df* = 12, *P*<0.05). There was no significant difference between the *kh$^w$* strain and controls with the black-blue, black-yellow and black-red groups (Fig 3E, t = -2.13, *df* = 12; t = -1.56, *df* = 8.71; t = -1.99, *df* = 12; *P*>0.05)

The mean values of *kh$^w$* strains flying into the three-hole Mosq-ovitrap under blue, green, and red lights were 0.33±0.21, 0.67±0.42 and 0.33±0.21, respectively, significantly differing from those of the control wild type (Fig 3F, t = 6.33, *df* = 5.38, *P*<0.001; t = 5.89, *df* = 11, *P*<0.001; t = 17.36, *df* = 6.76, *P*<0.001). The decreased number of mosquito traps *kh$^w$* to tricolor light suggested that *kh* was involved in the recognition of red, blue, and green monochromatic light or was related to the pattern formation of these three kinds of light.

## Discussion

This study reveals a pivotal role of compound eye-mediated vision in the search for oviposition sites using blindfolded and control mosquitoes. Additionally, the screening of differential genes related to vision and their functional analysis demonstrated that the *rho-l* and *kh* genes are crucial visual genes in the oviposition site searches of *Ae. albopictus*. These findings establish a foundation for the scientific control of *Ae. albopictus* breeding sites.

The ommatidium is the unit that makes up the compound eye and contains eight light-sensitive cells called retinular cells. The spectral sensitivity of these cells depends primarily on the opsin in each photoreceptor cell [53,54]. Opsins are generally divided into long wavelength-sensitive opsins, medium wavelength-sensitive opsins, and short wavelength-sensitive opsins [55].

This study shows that the expression of the long-wavelength opsin *rho-l* differs between the compound eye blindfolded group and the control group. The mosquito's perception of red light was reduced significantly after the ablation of *rho-l*, demonstrating that mosquitoes perceive long-wavelength light in dim light through the *rho-l* opsin, guiding them to search and locate suitable sites for oviposition. Results supports the interpretation that the *rho-l* gene is involved in the recognition of red light, which is consistent with the previous results of the two-color selection of the black-red containers, and that red light (most likely reflected in natural conditions) is important for oviposition site detection and selection [56]. Given the diversity of the opsin family, the co-expression of multiple opsins in the same photoreceptor cell enhances the sensitivity of photoreceptor cells to the light spectrum [57,58]. Hence, other long-wavelength opsins also may play a role in *Ae. albopictus*' search for egg-laying sites. At the same time, more than one opsin-related gene was screened in our transcriptome data, which can be further functionally analyzed.

Furthermore, pigment derived from tryptophan metabolism can regulate the luminous flux of compound eyes, form different color patterns, and absorb free radicals, thereby participating in cell homeostasis. The mosquito KMO gene was first identified in 1997 [59], and its metabolic pathway was determined by expression profiling and chemical reactions and suggested to be associated with ocular pigmentation in adult mosquitoes [60]. The *kh* gene encodes kynurenine hydroxylase, which hydrolyzes kynurenine to 3-HK (3-hydroxy-kynurenine), which is transferred to the compound eye to participate in the synthesis of pigment cells and protects the integrity of the retina by reducing the intensity of light reaching the photoreceptor and removing oxidants [41]. If the pigment cells are damaged, the retinular cells cannot

discern the intensity of light and this may result in damage to them [61]. Our experimental findings support the interpretation that the *kh* mutant strain exhibited a significantly diminished regulation of light intensity and was unable to perceive blue, green, and red light simultaneously.

The difference in oviposition activity index between the black-white and black-green groups supports the hypothesis that the pigment synthesized by *kh* gene might be involved in the recognition and permeability of green and white colors, and a *kh* knockout significantly enhanced the contrast between the white and green containers and the surrounding background, thus improving the recognition and selection of white and green oviposition containers. This study provides the initial evidence that the *Ae. albopictus kh* gene impacts adult female oviposition site selection. Furthermore, this functional defect may influence other physiological behaviors regulated by light pathways, such as host localization, mating, oviposition, and circadian rhythm. Considering that the *kh* gene serves as a common genetic marker for gene editing in insects [44,62,63], the fitness cost associated with $kh^w$ mutant strains, including visual impairment, warrants attention.

Blindfolding affected the ability of mosquitoes to identify preferred oviposition containers, but did not block completely their ability to do so. It is clear from a number of studies that olfactory cues also play a significant role in this behavior [8,11,12]. The results presented here show that although the *rho-l* and *kh* strains were significantly impaired in this behavior compared to wild-type controls in low-light conditions (0 lux), some mosquitoes had other senses, most likely, olfaction, guiding their behavior. Furthermore, the mutation of *kh* and *rho-l* genes also may lead to a decrease in locomotor activity, and this may account for the reduced numbers of mosquitoes trapped in the dark when compared to wild-type controls. The reduced searching ability of the $kh^w$ in dark environments may be due, in part, to its impact on mobility. Previous literature has reported that *kh* mutations in insects lead to motor dysfunction [64,65], which encompasses a range of behaviors, including flight. This is one possible explanation, another potential reason is that the *kh* gene may be involved in light recognition, requiring more time to adapt to changes in darkness-light transitions.

The observation that adult female *Ae. albopictus* prefer to find oviposition sites at dusk may indicate that twilight has specific attractive characteristics [46]. For example, this is a time when yellow light wavelengths are selectively absorbed by atmospheric ozone and the blue and red wavelengths will increase relative to each other, known as the "Chappuis effect" [66]. Additionally, longer red and orange light wavelengths are not easily scattered by the atmosphere, while shorter blue and violet light wavelengths are. As the sun approaches the horizon, atmospheric scattering by water vapor and dust becomes more pronounced, resulting in more scattering of blue and violet light and retention of red light. Consequently, the twilight spectrum is dominated by long wavelength red light, accompanied by low light intensity [67,68]. This study provided evidence that the *rho-l* gene product in the *Ae. albopictus* photoreceptor cells was sensitive to long wavelength red light, and the *kh* in the pigment cells was more adaptive to regulating the low light intensity at dusk and easily discerning the patterns formed by red light. This visual ability assists accurate identification of suitable oviposition sites at twilight.

The primary reason that *Ae. albopictus* females prefer to deposit their eggs in small black containers of stagnant water is that the black objects essentially do not reflect the spectrum, leading to decreased surface brightness, thus forming a strong contrast with the surrounding environment that attracts the mosquitoes [69,70]. The *kh* gene products result in the production of pigments that mainly mediate brightness perception. Mosquitoes with a mutant *kh* gene have a decreased ability to perceive brightness, resulting in increased visual contrast between white and green containers and their surroundings, thereby enhancing their recognition of these colored oviposition sites. However, *kh* knockout mutants may not perceive

significant differences in brightness for containers of other colors. Therefore, decreasing the amount of light that reaches the surface of the oviposition containers and increasing the visual contrast between the oviposition container and its surroundings may be an effective method for enhancing the induction of mosquito egg laying.

Large water bodies absorb more long wavelengths of light and reflect short wavelengths such as blue and green, while small water bodies reflect more long wavelength red light [71]. As a result, the red light emitted from small water bodies is more readily detected by the *rho-l* gene product in the photoreceptor cells and the *kh* downstream metabolites in pigment cells. This visual distinction may form the basis for the female's preference for depositing her eggs in small pools of water.

## Conclusions

This study demonstrates a role for vision in the oviposition site selection behaviour of adult female *Ae. albopictus*. It also provides evidence of a mechanistic role of the wild-type *rho-l* and *kh* gene products in this behavior with mutants exhibiting differential sensitivity or recognition of light intensity and substrate colors. This work may provide the basis for novel management practices for controlling *Ae. albopictus* abundance by manipulating oviposition sites.

## Supporting information

**S1 Fig. Comparison of the shadow index of blindfolded compound eyes with wild-type gravid mosquitoes.** Comparison of the shadow index of applied compound eyes with wild-type gravid mosquitoes. CE-blindfolded is the use of acrylic paint to apply mosquito compound eyes. shadow index = (number of mosquitoes in black environment—number of mosquitoes in light environment)/total. (t-test, t = 5.039, ***$P<0.001$, n = 9, Error bar represent mean ± SEM).
(TIF)

**S2 Fig. Activity monitoring of wild-type and *rho-l* mutant strains of *Aedes albopictus*.** (a) Line graphs of mean activity of *Aedes albopictus* wild-type strain and $rho\text{-}1^{\Delta807}$ strain under LD light conditions for 4 consecutive days (days 4 to 7), white bars represent daytime and black bars represent nighttime; (b) Comparison of daytime and nighttime activity of *Aedes albopictus* wild-type strain and $rho\text{-}1^{\Delta807}$ strain under LD light conditions, *Aedes albopictus* wild-type, n = 29. $rho\text{-}1^{\Delta807}$ strain, n = 28. (t-test, *$P<0.05$, **$P<0.01$, Error bar represent mean ± SEM).
(TIF)

**S3 Fig. Experimental Design Flowchart.**
(TIF)

**S1 Table. Amino acid sequences and base sequences of wild-type mosquitoes and mutant strains.**
(DOCX)

**S2 Table. List and sequence of oligonucleotide primers used in this study.**
(DOCX)

**S3 Table. Statistics of mutation rates of $G_0$ adults in *Ae. albopictus*.**
(DOCX)

**S4 Table. Reads of all detected genes in transcriptome sequencing.**
(XLS)

**S5 Table. The statistical data of the partial graph.**
(XLSX)

## Acknowledgments

The authors would like to thank the Guangdong Provincial Center for Disease Control and Prevention for providing the *Ae. albopictus* Foshan strain and other colleagues from Southern Medical University for their advice and assistance in this study.

## Author Contributions

**Conceptualization:** Si Yu Zhao, Xiao-Guang Chen.

**Data curation:** Si Yu Zhao, Hong Kai Liu.

**Formal analysis:** Si Yu Zhao, Hong Kai Liu, Zhen Sheng Xie.

**Funding acquisition:** Xiao-Guang Chen.

**Investigation:** Si Yu Zhao, Yi Ming Wu, Pei Lin Wu, JunYu Fu, Chun Mei Wang.

**Methodology:** Si Yu Zhao, Hong Kai Liu, Zhen Sheng Xie, Yi Ming Wu, Pei Lin Wu, Tong Liu, Wen Qiang Yang.

**Project administration:** Xiao-Guang Chen.

**Resources:** Xiao-Guang Chen.

**Software:** Si Yu Zhao, Hong Kai Liu, Jing Wu.

**Supervision:** Anthony A. James, Xiao-Guang Chen.

**Validation:** Hong Kai Liu, Tong Liu, Wen Qiang Yang.

**Visualization:** Si Yu Zhao, Hong Kai Liu, Zhen Sheng Xie.

**Writing – original draft:** Si Yu Zhao, Xiao-Guang Chen.

**Writing – review & editing:** Si Yu Zhao, Hong Kai Liu, Anthony A. James, Xiao-Guang Chen.

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
