## [Decision Letter · Decision Letter 0]

17 Sep 2024

Dear Professor Chen,

Thank you very much for submitting your manuscript "Vision guides the twilight search for oviposition sites of the Asian tiger mosquito, Aedes albopictus" for consideration at PLOS Neglected Tropical Diseases. As with all papers reviewed by the journal, your manuscript was reviewed by members of the editorial board and by several independent reviewers. In light of the reviews (below this email), we would like to invite the resubmission of a significantly-revised version that takes into account the reviewers' comments. 

The reviewers indicated that your paper was interesting, but had some suggestions for improvement. After these are addressed, two or the reviewers requested to review that paper once more.

We cannot make any decision about publication until we have seen the revised manuscript and your response to the reviewers' comments. Your revised manuscript is also likely to be sent to reviewers for further evaluation.

Sincerely,

Joshua B. Benoit

Academic Editor

Amy Morrison

Section Editor

The reviewers indicated that your paper was interesting, but had some suggestions for improvement. After these are addressed, two or the reviewers requested to review that paper once more.

Reviewer's Responses to Questions

**Key Review Criteria Required for Acceptance?**

**Methods**

-Are the objectives of the study clearly articulated with a clear testable hypothesis stated?

-Is the study design appropriate to address the stated objectives?

-Is the population clearly described and appropriate for the hypothesis being tested?

-Is the sample size sufficient to ensure adequate power to address the hypothesis being tested?

-Were correct statistical analysis used to support conclusions?

-Are there concerns about ethical or regulatory requirements being met?

Reviewer #1: The objectives of the study clearly articulated, and the study design was appropriate.

Reviewer #2: No problems for the methods section.

Reviewer #3: The only methodological modification I am suggesting is the inclusion of other controls for the RNA-seq experiments i.e. blindfolded compounded eyes not flying into the three-hole Mosq-ovitrap and unblindfolded compound eyes not flying into the three-hole Mosq-ovitrap.

**Results**

-Does the analysis presented match the analysis plan?

-Are the results clearly and completely presented?

-Are the figures (Tables, Images) of sufficient quality for clarity?

Reviewer #1: Statistical analysis was clear, and results were completely presented.

Reviewer #2: Confident results are presented.

Reviewer #3: (No Response)

**Conclusions**

-Are the conclusions supported by the data presented?

-Are the limitations of analysis clearly described?

-Do the authors discuss how these data can be helpful to advance our understanding of the topic under study?

-Is public health relevance addressed?

Reviewer #1: The conclusion supported by the results presented. the discussion was helpful.

Reviewer #2: A clear conclusion is drawn.

Reviewer #3: (No Response)

**Editorial and Data Presentation Modifications?**

Reviewer #1: (No Response)

Reviewer #2: Minor revision.

Reviewer #3: (No Response)

**Summary and General Comments**

Reviewer #1: The authors attempted to prove that mosquito’s vision is important in their search for oviposition habitats, and they used novel methods proved that some visual-related genes control the mosquito oviposition habitat search behavior. The experimental designs are appropriate, and the results are very interesting. 

Major comment

The major issue is the mix-up of results with methods and discussions. The authors described in several places the “purposes and methods” in the “result” section; whereas, in the “methods” section, the authors did not describe the purpose of each experiment. Similarly, after describing the results, the authors discussed them in the “results” section, which should be discussed in the “discussion” section.

Specific comments

1. Transcriptome sequencing, the purpose of this is not clear, to identify differentially expressed visual–related genes or to examine odor-related genes? The purpose of this experiment will determine the control and test groups. The purpose of this experiment needs to be clarified.

2. Transcriptome sequencing: “Mosquitoes with blindfolded compound eyes flying into the three-hole Mosq-ovitrap will be the test group, while those with unblindfolded eyes flying into the three-hole Mosq-ovitrap device will be the control group”, is this correct? Or should it be “Mosquitoes with blindfolded compound eyes (no visual cue) did not fly into the traps will be the test group, while those with unblindfolded eyes (has visual cue) flying into the traps will be the control group” if testing for visual-related genes? 

3. Similarly, the authors should describe the purpose of “CRISPR/Cas9 editing experiments of rho-l and kh genes” and for “Color bidirectional selection experiments”, so that readers have a better understanding of the context and to determine if the methods used are appropriate. In fact, the best way to do is to draw a flowchart to describe the flows of experimental design and the purpose of each experiments, although most studies lack this. 

4. Line 254: “The oviposition activity index is the number of eggs of other-colored oviposition mosquitoes - number of black oviposition mosquitoes)/ (number of eggs of other colored oviposition mosquitoes + number of black oviposition mosquitoes)”, why not simply use number of eggs of other-colored oviposition mosquitoes vs. number of black oviposition mosquitoes?

5. Line 301: “Statistical analysis”, “T-test” should be “t-test”, because usually “T-test” is used for “Holling T-test”

6. Lines 311-317: This section should move to “Method” section, this describes the purpose of the method used especially the authors include the references here. In the “Result” section, only describe the results. Lines 331-332 may also be moved to “Method” section.

7. In line 327, please use (b) and (d) instead of “b and d”. 

8. Lines 333-334: A statistical test result should be added here to describe the significance.

9. Lines 337-345: Similarly, this part should be described in the “Method” section, i.e., the purpose of each method. 

10. Line 348: need a degree of freedom for the t-test. This occurred in many places in the results, please check all t-test results in the text, please use “t=???, d.f. = ?, P-value”, all tests need a degree of freedom.

11. Lines 357-359: Please move this sentence to “Method” section.

12. Line 362: Need test method, statistical value, degree of freedom.

13. Lines 367-373: Please move this section to “Methods” section.

14. Line 408: Fig 2, please use “(d), (e), (f), and (g)”.

15. Lines 411-425: Should this part be moved to method? It described the method rather than results.

16. Lines 435-443: Please move this part to “Methods” section.

17. Lines 469-476: Please move this part to " Methods” section.

18. Lines 494-500: Please move this part to " Methods” section.

19. Lines 509-518: This section should be described in the “Discussion” section. 

20. Lines 512-526: This belongs to the “Method” section.

21. Lines 572-580: This section should be described in the “Discussion” section. 

22. Lines 588-592: This is the conclusion of the study, should be described at the end of the paper.

Reviewer #2: The work is of high scientific value and rigor. 

1. The authors say they used a brush to coat the mosquito's compound eyes with a black water-soluble pigment, but they don't say how much was used. This should be specified. 

2. Line 103 needs more literature support.

3.Line 358, change “should result in” to “will result in”.

4. Line 525, change “forms different color patterns” to “and forms different color patterns”.

5. Line 657 needs more literature support.

Reviewer #3: The manuscript entitled; “Vision guides the twilight search for oviposition sites of the Asian tiger mosquito, Aedes albopictus” provides a robust investigation into the understanding of the roles of visual cues in mosquito reproductive biology. The authors through a series of experiments ranging from behavioral, RNA-seq to gene-editing assays have been able to dive into some of the mechanistic factors influencing oviposition site selection in Aedes albopictus. The results here are important since the manipulation of oviposition sites can be leveraged for mosquito control. The paper is well written except for some minor editing and paragraphing issues which I will be highlighting subsequently. The results are well described and the appropriate tests were selected accordingly. The only big limitation in this work is the inability of the authors to conduct specific experiments where the influence of olfaction cues is disentangled from visual cues.

Introduction:

~ Lines 76-78: The sentence here seems incomplete if the public health importance of these diseases are not emphasized.

~ Lines 98-102: To improve readability, this sentence can be broken into two sentences.

Methods:

~ Generally, apart from the locomotor activity section, no information was given on the age of the mosquitoes when they were used for the experiments. The only thing close was the days after blood-feeding that the mosquitoes were selected. Age seems to play a very significant role on several aspects of mosquito biology, so it would be important if the age of the mosquitoes were mentioned.

~ For the RNA-seq experiments (lines 168-170), was there any reason these controls were not considered?: (a) blindfolded compound eyes not flying into the three-hole Mosq-ovitrap and (b) unblindfolded compound eyes not flying into the three-hole Mosq-ovitrap device.

~ Lines 254-258 - The equation is not easily comprehensible.

~ Lines 289-290 - The activity monitoring system has also been used in mosquitoes recently in the last five years: Ajayi et al 2022 (https://doi.org/10.1242/jeb.244032).

~ To address the issue of olfaction obscuring the effect of visual cues, the authors can do well by running specific experiments with mosquitoes showing mutations in olfactory responses (orco mutants etc).

Results:

~ SWISS-MODEL was mentioned once and there is was prior description about it. It will be helpful if you can provide details about the model whether in the methods or results section.

Discussion:

~ Lines 638-640 seems to be too short to be a paragraph.

PLOS authors have the option to publish the peer review history of their article (what does this mean?). If published, this will include your full peer review and any attached files.

Reviewer #1: No

Reviewer #2: No

Reviewer #3: No
---

## [Decision Letter · Decision Letter 1]

4 Nov 2024

Dear Professor Chen,

We are pleased to inform you that your manuscript 'Vision guides the twilight search for oviposition sites of the Asian tiger mosquito, Aedes albopictus' has been provisionally accepted for publication in PLOS Neglected Tropical Diseases.

Best regards,

Joshua B. Benoit

Academic Editor

Amy Morrison

Section Editor

Shaden Kamhawi

co-Editor-in-Chief

Paul Brindley

co-Editor-in-Chief

Reviewer's Responses to Questions

**Key Review Criteria Required for Acceptance?**

**Methods**

-Are the objectives of the study clearly articulated with a clear testable hypothesis stated?

-Is the study design appropriate to address the stated objectives?

-Is the population clearly described and appropriate for the hypothesis being tested?

-Is the sample size sufficient to ensure adequate power to address the hypothesis being tested?

-Were correct statistical analysis used to support conclusions?

-Are there concerns about ethical or regulatory requirements being met?

Reviewer #2: (No Response)

Reviewer #3: Yes

**Results**

-Does the analysis presented match the analysis plan?

-Are the results clearly and completely presented?

-Are the figures (Tables, Images) of sufficient quality for clarity?

Reviewer #2: (No Response)

Reviewer #3: Yes

**Conclusions**

-Are the conclusions supported by the data presented?

-Are the limitations of analysis clearly described?

-Do the authors discuss how these data can be helpful to advance our understanding of the topic under study?

-Is public health relevance addressed?

Reviewer #2: (No Response)

Reviewer #3: Yes

**Editorial and Data Presentation Modifications?**

Reviewer #2: (No Response)

Reviewer #3: (No Response)

**Summary and General Comments**

Reviewer #2: (No Response)

Reviewer #3: The authors have sufficiently provided clarifications to all my concerns.

PLOS authors have the option to publish the peer review history of their article (what does this mean?). If published, this will include your full peer review and any attached files.

Reviewer #2: No

Reviewer #3: No

---

## [Editor Report · Acceptance letter]

11 Nov 2024

Dear Professor Chen,

We are delighted to inform you that your manuscript, "Vision guides the twilight search for oviposition sites of the Asian tiger mosquito, Aedes albopictus," has been formally accepted for publication in PLOS Neglected Tropical Diseases.

Best regards,

Shaden Kamhawi

co-Editor-in-Chief

Paul Brindley

co-Editor-in-Chief
